# The alternative cap-binding complex is required for antiviral defense *in vivo*

Anna Gebhardt[1,2], Valter Bergant[1,2], Daniel Schnepf[3,4], Markus Moser[5,6], Arno Meiler[2], Dieudonnée Togbe[7], Claire Mackowiak[7], Line S. Reinert[8], Søren R. Paludan[8], Bernhard Ryffel[7,9], Alexey Stukalov[1,2], Peter Staeheli[3,10], Andreas Pichlmair[1,2,11] *

**1** Institute of Virology, Technical University of Munich, School of Medicine, Munich, Germany, **2** Innate Immunity Laboratory, Max-Planck Institute of Biochemistry, Martinsried, Germany, **3** Institute of Virology, University of Freiburg, Freiburg, Germany, **4** Spemann Graduate School of Biology and Medicine, Albert Ludwigs University Freiburg, Freiburg, Germany, **5** Department of Molecular Medicine, Max-Planck Institute of Biochemistry, Martinsried, Germany, **6** Center for Translational Cancer Research (TranslaTUM), TUM School of Medicine, Technical University of Munich, Munich, Germany, **7** INEM, Experimental Molecular Immunology, UMR7355 CNRS and University, Orleans, France, **8** Department of Biomedicine, University of Aarhus, Aarhus, Denmark, **9** Institute of Infectious Diseases and Molecular Medicine, University of Cape Town, Cape Town, South Africa, **10** Faculty of Medicine, University of Freiburg, Freiburg, Germany, **11** German Center for Infection Research (DZIF), Munich partner site, Munich, Germany

* andreas.pichlmair@tum.de

**Data Availability Statement:** All relevant data are within the manuscript and its Supporting Information files.

**Funding:** This work was supported by the Max-Planck Free Floater program to A.P., the German

## Abstract

Cellular response to environmental challenges requires immediate and precise regulation of transcriptional programs. During viral infections, this includes the expression of antiviral genes that are essential to combat the pathogen. Transcribed mRNAs are bound and escorted to the cytoplasm by the cap-binding complex (CBC). We recently identified a protein complex consisting of NCBP1 and NCBP3 that, under physiological conditions, has redundant function to the canonical CBC, consisting of NCBP1 and NCBP2. Here, we provide evidence that NCBP3 is essential to mount a precise and appropriate antiviral response. Ncbp3-deficient cells allow higher virus growth and elicit a reduced antiviral response, a defect happening on post-transcriptional level. Ncbp3-deficient mice suffered from severe lung pathology and increased morbidity after influenza A virus challenge. While NCBP3 appeared to be particularly important during viral infections, it may be more broadly involved to ensure proper protein expression.

## Author summary

Infection with viruses and other pathogens requires appropriate cellular countermeasures, which involve swift and accurate adaptation of gene expression profiles. mRNAs encoding for immune-regulatory and effector proteins need to be transported into the cytoplasm in order to generate proteins necessary to fight the pathogen. Here we show that this process requires proper functionality of the Nuclear cap protein 3 (NCBP3), a protein recently identified to contribute to an alternative mRNA cap-binding complex. An Ncbp3-deficient mouse model allowed higher virus growth *in vitro* and showed high susceptibility to influenza A virus challenge *in vivo*. While NCBP3-deficient cells were able to transcriptionally

research foundation to A.P. (PI1084/3 and TRR179 TP11) and P.S. (SFB 1160, project 13), the Lunbeck Foundation to L.R. (R198 2015 171), the Centre National de la Recherche Scientifique (CNRS), University and European Regional Development Fund (FEDER (N° 2016-00110366 and EX005756) to B.R., an ERC consolidator grant to A.P (ERC CoG ProDAP, 817798) and the Infect-Era and the German Federal Ministry of Education and Research (ERASE) to A.P. The funders had no role in study design, data collection and analysis, decision to publish, or preparation of the manuscript.

**Competing interests:** The authors have declared that no competing interests exist.

upregulate cytokine mRNAs, generation of cytokines was significantly reduced in the absence of NCBP3. Our data shows a non-redundant function of NCBP3 and the alternative cap-binding complex in antiviral responses. More broadly, this work demonstrates a yet unappreciated aspect of post-transcriptional gene regulation.

## Introduction

Successful control of virus infections requires immediate and appropriate response to invading pathogens, a process coordinated by the innate immune system. The activation of the innate immune system is predominantly initiated by viral nucleic acids that are delivered into the cell during the infection process and bear virus-specific chemical or structural signatures [1]. Pathogen-associated molecular patterns (PAMPs) are sensed by specific pattern recognition receptors (PRRs), which activate signaling cascades leading to transcription of antiviral defense genes. This includes antiviral and inflammatory cytokines such as type I interferons (IFN-α/β), tumor necrosis factor alpha (TNF-α) and interleukins (IL) including pro-inflammatory IL-6, IL-8 and IL-12 [2]. Cytokines act in a paracrine and autocrine manner and bind to specific receptors, which trigger secondary signaling cascades leading to transcriptional activation of antiviral and inflammatory effector genes. This includes several hundred IFN-stimulated genes (ISGs) such as interferon-induced proteins with tetratricopeptide repeats (IFITs), orthomyxovirus resistance genes (MX) or 2'-5'-oligoadenylate synthases (OAS) [3, 4]. Transcripts are exported from the nucleus to the cytoplasm and translated into bioactive proteins that form a large antiviral network [5]. Transcription, mRNA processing, export and translation are therefore indispensable to effectively counteract virus infection and to mount an appropriate innate immune response.

To protect cellular RNA polymerase II transcripts from degradation and to guide them through the sequence of steps leading from transcription to translation, the transcripts, including messenger RNAs (mRNAs), are co-transcriptionally capped at the 5'-end by a mono-methylated guanosine through a 5'-5' triphosphate linkage (cap) [6]. The RNA cap is bound by the cap-binding complex (CBC), which coordinates RNA processing, export and initiation of translation. The CBC is a heterodimeric complex consisting of nuclear cap binding protein 1 (NCBP1) and NCBP2 [7]. While NCBP2 directly binds the cap, NCBP1 allows recruitment of cellular factors required for subsequent mRNA maturation [6, 7, 8]. After capping, RNA molecules undergo a series of modifications, most notably splicing and 3'-end polyadenylation, which are tightly controlled by multiple protein complexes. It was shown that accurate mRNA processing allows its association to the export receptor NXF1-NXT1 (TAP-p15) and is a prerequisite for efficient export through the nuclear pore complex (NPC) [9].

We recently identified an alternative CBC consisting of the cap-binding protein NCBP3 and NCBP1 [10]. Under physiological conditions the canonical and alternative CBCs show a high degree of functional redundancy in facilitating export of RNA polymerase II transcripts, with the alternative CBC showing increased affinity to mRNA than to the other types of RNA. We have previously shown that under physiological conditions, depletion of either NCBP2 or NCBP3 does not affect cell viability, while depletion of both cap-binding proteins leads to reduced cell growth. Similarly, only co-depletion of NCBP2 and -3 or the sole common adapter NCBP1 retains polyadenylated RNA in the nucleus. Why eukaryotes evolved two CBCs, consisting of distinct cap-binding proteins bound to a common adaptor, remained unclear. We speculated that the alternative CBC may be particularly important in situations of cellular stress such as during virus infections.

Viruses have identified mRNA processing as a primary target for downregulating immune responses and supporting viral replication and spread. Influenza A virus (IAV), for instance, encodes the nonstructural protein 1 (NS1) to counteract and downregulate host innate immune responses, in part through targeting multiple steps in the mRNA processing and nuclear export [11, 12]: (I) IAV-NS1 blocks RNA 3'-end processing through inhibiting components of the cleavage and polyadenylation specificity factor (CPSF) complex. (II) IAV-NS1 binds directly to PABPN and thereby pertains proper mRNA polyadenylation and transport to the cytoplasm. (III) IAV-NS1 associates to the export receptor NXF1-NXT1 and the nuclear pore protein NUP98. Another example of mRNA export manipulation is the vesicular stomatitis virus (VSV) matrix protein (M), which attaches to the RNA binding protein RAE1 and prevents its interaction with nucleoporin NUP98, leading to accumulation of mRNA and small nuclear RNA (snRNA) in the nucleus [12]. Impaired mRNA processing and export inhibit appropriate immune responses and therefore paralyze cells in a vulnerable state. Conversely, mutations in IAV-NS1 or VSV-M proteins lead to attenuation of these virus strains and even allow their usage as safe therapeutic agents [13, 14].

The recently identified alternative CBC consisting of NCBP1 and NCBP3 under physiological conditions functions in a redundant manner to the canonical CBC consisting of NCBP1 and NCBP2. However, NCBP3 was required to limit virus growth *in vitro*. The functional basis for this phenomenon remained unclear as well as its relevance for the immune response *in vivo*. Here, we show that NCBP3 is required to facilitate translation of cytokine mRNA into functional proteins and to mount a proper immune response during virus infections *in vitro*. Genetic depletion of *Ncbp3 in vivo* rendered mice susceptible to IAV infection, indicating a critical role of NCBP3 during antiviral responses *in vivo*.

## Results

### Generation and characterization of *Ncbp3* knockout mice

To test the function of NCBP3 *in vivo*, we generated *Ncbp3* knockout (ko) mice from embryonic stem cells with an inserted knockout cassette in the *Ncbp3* gene locus (S1A Fig). Correct insertion of the knockout cassette was confirmed by gene-specific PCR assays and resulted in 49% of heterozygous F0 mice (S1A and S1B Fig). Heterozygous *Ncbp3* knockout mice (*Ncbp3*$^{+/-}$) showed no phenotype as compared to littermate controls. However, when breeding *Ncbp3*$^{+/-}$ mice we obtained reduced numbers of *Ncbp3*-deficient (*Ncbp3*$^{-/-}$) offspring (Fig 1A, S1D Fig). Similarly, reduced number of homozygous mice was obtained upon breeding *Ncbp3*$^{-/-}$ to heterozygous animals (Fig 1A). Born *Ncbp3*$^{-/-}$ mice showed reduced body weight compared to littermate controls but no other obvious phenotype during an observation period of 14 months (S1E and S1F Fig). Tissues of 12–13 weeks-old mice were analyzed for *Ncbp3* promoter activity by quantifying LacZ expression under the control of the endogenous *Ncbp3* gene regulatory elements (Fig 1B). Moderate expression was observed in bone marrow, pancreas, spleen, thymus, sexual organs, kidney, and liver, high expression in lungs and brain, and highest expression in heart and muscle, with minor differences between male and female animals.

*Ncbp3* wild-type (*Ncbp3*$^{+/+}$) and *Ncbp3*$^{-/-}$ mouse embryonic fibroblasts (MEFs) isolated from heterozygous crossings were morphologically similar. Loss of *Ncbp3* was confirmed by western blot analysis employing a NCBP3-specific antibody (Fig 1C). Heterozygous MEFs showed intermediate expression of NCBP3. We proceeded to use these cells to investigate the function of the alternative CBC *in vitro*. To test whether the canonical CBC is operative in murine cells, we employed a transient knockdown experiment depleting NCBP1 and NCBP2 by siRNA. Knockdown efficiency was confirmed by western blot analysis (Fig 1D). In line with

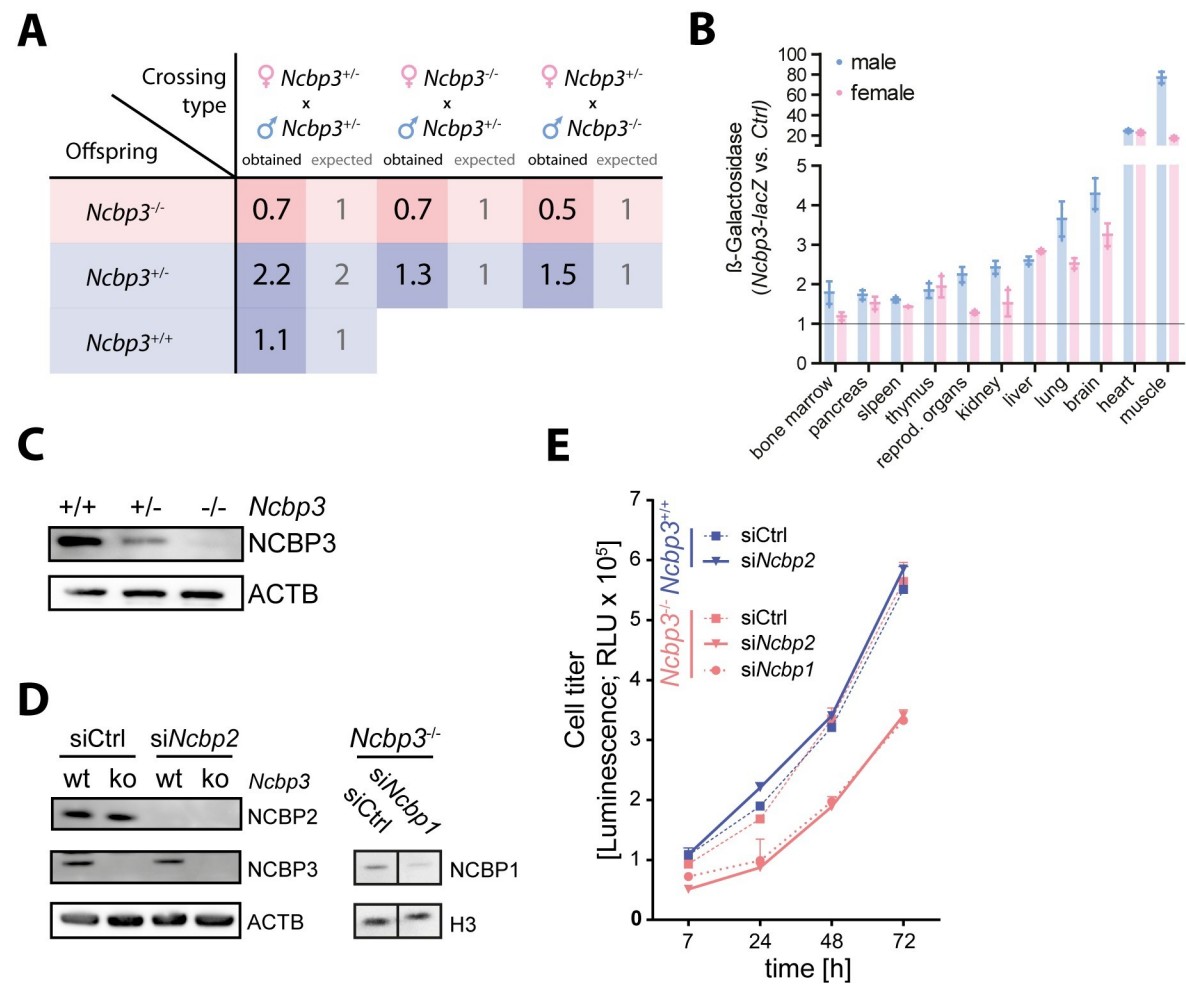

**Fig 1. Characterization of *Ncbp3* knockout mice.** (A) Mendelian frequency of *Ncbp3 tm1a* crossings. Expected and obtained Mendelian ratios of *F1 Ncbp3 tm1a* crosses are presented. (B) Expression of *Ncbp3* in different tissues. Tissues from 3 animals per genotype and gender were isolated and individual and mean values ± SD normalized to background signal are presented. (C) Western blot analysis of *Ncbp3 tm1a* MEFs. MEFs were generated from *Ncbp3* wild-type (wt; *Ncbp3^{+/+}, tm1a^{-/-}*), heterozygous (het; *Ncbp3^{+/-}, tm1a^{+/-}*) and knockout (ko; *Ncbp3^{-/-}, tm1a^{+/+}*) embryos and western blot analysis was performed using antibodies against indicated proteins. (D) Knockdown efficiency of NCBP1 and NCBP2 was confirmed by western blot, staining with antibodies against indicated proteins. (E) Cell growth of *Ncbp3^{+/+}* and *Ncbp3^{-/-}* MEFs after RNAi-mediated knockdown. MEFs were electroporated twice with siRNAs against *Ncbp1*, *Ncbp2* or non-targeting siRNA (siCtrl). Cell viability was measured as described in Materials and Methods at indicated time points. RNAi treatment was performed in triplicates and the graph displays the mean ± SD. NCBP1/2/3, Nuclear cap-binding protein 1/2/3; wt, wild-type; het, heterozygous; ko, knockout; RLU, relative light units.

results in human cells [10], NCBP2 depletion did not affect growth of *Ncbp3^{+/+}* MEFs compared to control siRNA treatment (Fig 1E). In contrast, knockdown of NCBP2 significantly reduced growth of *Ncbp3^{-/-}* MEFs. This effect was comparable to reduction of cell growth after NCBP1 depletion. These experiments indicate that the function of the canonical CBC (NCBP1- NCBP2) and the alternative CBC (NCBP1- NCBP3) is conserved in murine and human cells. It also supports the notion that the two CBCs duplicate the function of each other under normal physiological conditions.

## Loss of NCBP3 increases RNA virus replication *in vitro*

A phenotype of NCBP3 depletion was previously shown to become apparent under challenging environmental conditions [10]. We employed viral infection, a highly stressful condition requiring a prompt cellular response, to test for the specific role of NCBP3. To this aim we selected influenza A virus (IAV) and vesicular stomatitis virus (VSV), which replicate in the nucleus and cytoplasm, respectively. In addition, we used viral mutants that lost their ability to efficiently inhibit the innate immune response (IAV-ΔNS1, VSV-M2). Notably, infection with different doses of an IAV reporter virus expressing renilla luciferase showed that *Ncbp3*$^{-/-}$ MEFs allowed increased IAV-renilla reporter activity as compared to *Ncbp3*$^{+/+}$ MEFs (Fig 2A). We reconstituted *Ncbp3*$^{-/-}$ MEFs with control or *Ncbp3*-encoding lentiviruses that allow doxycycline-dependent transgene induction. Transient expression of NCBP3 in *Ncbp3*$^{-/-}$ MEFs reduced IAV-renilla reporter activity to similar extent as seen when comparing *Ncbp3* wt and -ko MEFs (Fig 2A–2C), confirming a direct effect of *Ncbp3*. To further corroborate the increased susceptibility of *Ncbp3*$^{-/-}$ MEFs towards IAV, we infected cells with IAV-wt and quantified infectious viral particles in the supernatants. *Ncbp3*$^{-/-}$ MEFs produced 30-fold more infectious IAV particles compared to *Ncbp3*$^{+/+}$ controls (Fig 2D). The IFN-inducing variant IAV-ΔNS1 also grew to over 10-fold higher titers in *Ncbp3*$^{-/-}$ as compared to *Ncbp3*$^{+/+}$ MEFs (Fig 2D).

In order to further corroborate observed increased susceptibility of *Ncbp3*$^{-/-}$ MEFs towards IAV, we performed western blots to analyze viral protein accumulation and explored the activation of antiviral response in terms of expression of interferon inducible protein IFIT1. In line with a role in virus growth regulation, accumulation of the viral nucleoprotein (NP) and non-structural protein 1 (NS1) was increased in *Ncbp3*$^{-/-}$ MEFs as compared to *Ncbp3*$^{+/+}$ counterparts (Fig 2E). Despite the higher accumulation of viral proteins, expression of IFIT1 in response to virus infection was similar for both genotypes (Fig 2E). The discrepancy between viral and IFIT1 protein abundance was especially apparent upon infection with IAV-ΔNS1. This phenotype is not in line with an expected correlation between the levels of viral and antiviral defense proteins and is suggestive for an involvement of NCBP3 in mounting an appropriate immune response.

Similar results were obtained when assessing the abundance of viral RNA in infected cells. IAV RNA was present at 10-fold higher levels in *Ncbp3*$^{-/-}$ MEFs as compared to wild-type controls (Fig 2F). Higher accumulation of viral mRNA was accompanied by higher interferon beta (*Ifnb1*) and tumor necrosis factor (*Tnf*) mRNA levels in *Ncbp3* ko cells as compared to wt controls, indicating proper activation of interferon regulatory factor (IRF) and nuclear factor kappa B (NF-κB) signaling in *Ncbp3* ko cells (Fig 2F). Surprisingly, despite higher viral and *Ifnb1* mRNA expression in *Ncbp3*-deficient as compared to wt MEFs, accumulation of interferon-induced *Ifit1* and *Ifit3* mRNA was comparable in both cell types (Fig 2F), further indicative of the imbalance in induction of an antiviral response in *Ncbp3* ko MEFs. RNA of the housekeeping gene *Hmbs* was similarly expressed upon infection and in *Ncbp3*-ko compared to wt cells (Fig 2F), supporting our previous observations that generic mRNA maturation is not affected by the absence of NCBP3.

In order to further study viral activation of innate immune pathways potentially dependent on *Ncbp3*, we quantified amounts of secreted IRF- and NF-κB-regulated cytokines IP-10 and IL-6 upon infection of MEFs with IAV-ΔNS1. IP-10 secretion was not significantly different between *Ncbp3*$^{+/+}$ and *Ncbp3*$^{-/-}$ MEFs (Fig 2G), further corroborating the lack of ISG response of appropriate magnitude relative to viral replication levels (Fig 2D). Interestingly, IL-6 secretion was significantly attenuated in *Ncbp3* ko as compared to wt MEFs (Fig 2G), highlighting an imbalance between NF-κB signaling pathway activation and corresponding cytokine secretion in *Ncbp3* ko MEFs.

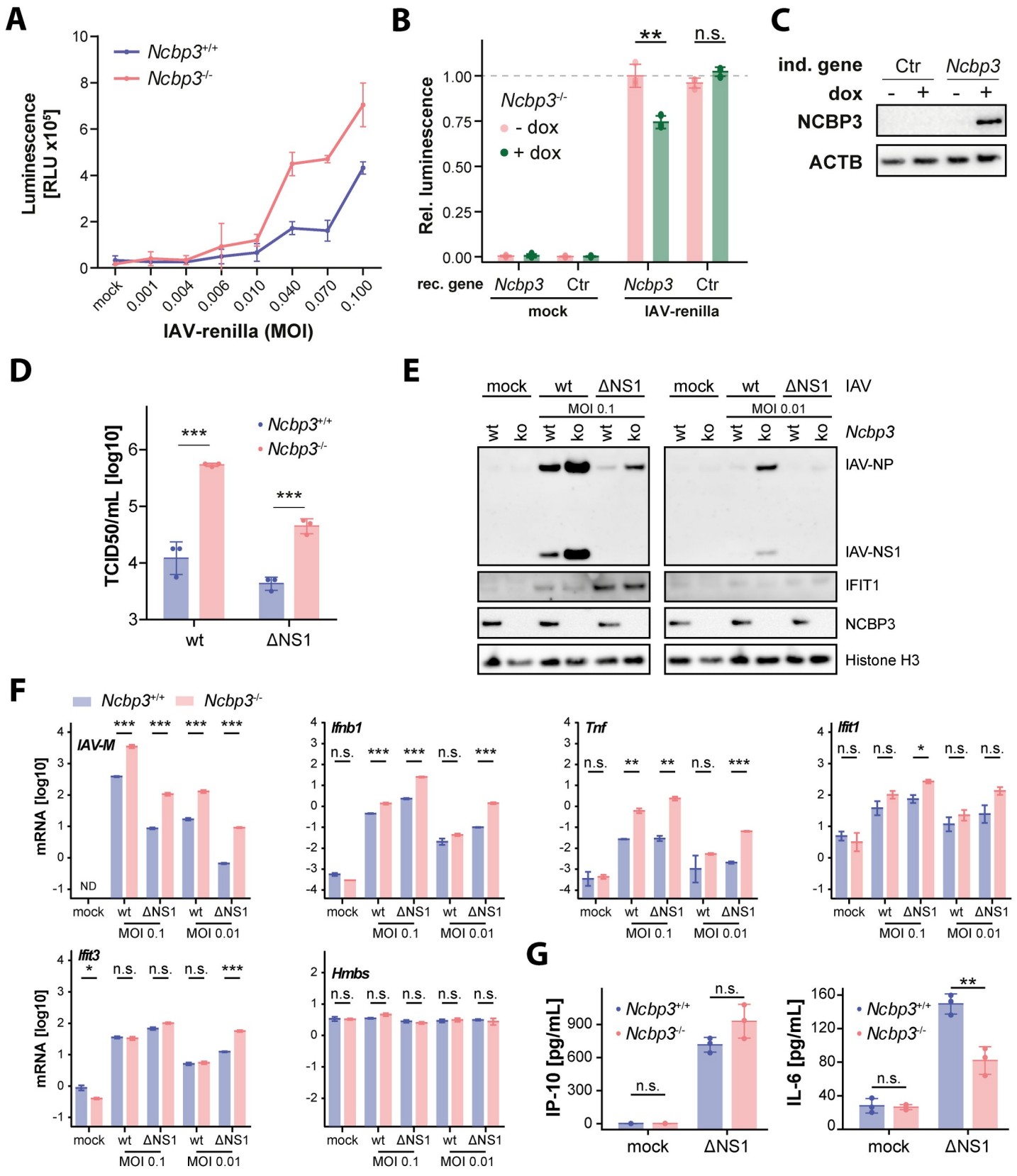

**Fig 2. Loss of NCBP3 supports increased IAV replication *in vitro*.** (A) Luciferase activity in lysate of *Ncbp3* wt and ko MEFs upon infection at different MOIs. MEFs were infected with an IAV-wt reporter virus (strain SC35M) expressing renilla luciferase using indicated MOIs for 21 hours. Luminescence counts were normalized to total protein concentration. Graphs display the mean ± SD of virus infections performed in triplicates. (B) Relative luciferase activity in lysate of *Ncbp3* ko MEFs transduced with doxycycline-inducible *Ncbp3* or no-transgene (Ctr) encoding lentiviruses and infected with IAV-renilla reporter virus for 21 hours. Graph displays mean ± SD of three measurements. Representative of three independent repeats is shown. Rec. gene, reconstituted gene, n.s. *P>0.05*, ** *P<0.01* as analyzed by two-tailed equal variance t-test. (C) Induction of reconstituted gene was confirmed by western blot using antibodies against indicated proteins. (D) Accumulation of infectious viral particles in supernatants of IAV-infected MEFs quantified by TCID50. MEFs were infected with IAV-wt and -ΔNS1 (strain SC35M) using an MOI of 0.01 and 0.1, respectively, for 24 hours. *** *P<0.001* as analyzed by two-way analysis of variance (ANOVA) with Bonferroni's p-value correction. (E) Western blot analysis of IAV-infected *Ncbp3* wt and ko MEFs using antibodies against indicated proteins. MOI, multiplicity of infection; IAV, Influenza A virus; IFIT1, Interferon-induced protein with tetratricopeptide repeats 1; IAV-NP, Influenza A virus nucleoprotein; IAV-NS1, Influenza A virus non-structural protein 1. (F) mRNA expression of indicated genes in IAV-infected *Ncbp3* wt and ko MEFs. MEFs were infected with indicated MOIs of IAV-wt and -ΔNS1 for 24 hours and transcript abundance analyzed by qRT-PCR. Data were normalized to murine *Tbp* mRNA. The mean of two technical replicates of a representative of four separate infection experiments is shown. N.s. *P>0.05*, * *P<0.05*, ** *P<0.01*, *** *P<0.001* as analyzed by two-tailed equal variance t-test. *IAV-M*, Influenza A virus matrix protein; *Ifnb1*, Interferon beta; *Tnf*, tumor necrosis factor; *Hmbs*, hydroxymethylbilane synthase; ND, not detected. (G) Concentrations of secreted cytokines in MEFs infected with IAV-ΔNS1 at an MOI of 0.1 for 24 hours as determined by ELISA against indicated proteins. Graph displays mean ± SD of three separate infection experiments. N.s. *P>0.05*, ** *P<0.01* as determined by two-tailed equal variance t-test.

In sum, these experiments indicate that NCBP3 is required to control IAV infection *in vitro*. Despite significantly higher accumulation of viral proteins and viral nucleic acids in *Ncbp3⁻/⁻* cells, the induction of cytokines and antiviral response genes was equal or even reduced in *Ncbp3*-ko MEFs as compared to wt controls, indicating a potential inability to mount an appropriate innate immune response in the absence of NCBP3.

To exclude that the effect of NCBP3 depletion is specific to IAV, we also tested growth of VSV, a virus that replicates in the cytoplasm. In line with data obtained for IAV, VSV-driven luciferase signal was increased in *Ncbp3⁻/⁻* MEFs as compared to wt controls (Fig 3A). Accumulation of infectious virus in supernatants of *Ncbp3⁻/⁻* relative to *Ncbp3⁺/⁺* MEFs was 100-fold increased for VSV-wt as well as the interferon-inducing variant VSV-M2 (Fig 3B). Notably, despite the significantly increased virus growth, expression of IFIT1 protein was again similar in *Ncbp3*-wt and -ko MEFs (Fig 3C). These results indicate that *Ncbp3*-deficient MEFs cannot support an innate immune response to an extent required to cope with the increased viral load. Altogether, these data show that infection with unrelated RNA viruses, regardless of their replication site, results in higher accumulation of viral particles in supernatant of NCBP3-depleted cells. Activation of the innate immune responses is similar in *Ncbp3*-wt and -ko MEFs but execution of the innate immune response was impaired in *Ncbp3*-depleted cells.

## Global proteomic analysis suggests altered innate immune response of a subset of ISGs in *Ncbp3*-deficient MEFs

Our data indicate that *Ncbp3*-wt and -ko MEFs differ in their ability to translate innate immune signals into adequate response on the level of mRNA and proteins. In order to investigate this defect on a proteome-wide scale we performed proteomic analysis of IAV-infected MEFs. Briefly, MEFs were infected with IAV-wt for 24 hours and total viral and host protein expression was analyzed by liquid chromatography coupled to tandem mass spectrometry (LC-MS/MS) (Fig 4A). This analysis allowed quantification of 5422 proteins in *Ncbp3⁺/⁺* and *Ncbp3⁻/⁻* MEFs (S1 Table). In line with the increased virus growth, we could observe increased expression of viral proteins in *Ncbp3*-deficient as compared to wt MEFs (Fig 4B). Despite increased accumulation of viral proteins, bulk ISG-expression was reduced upon IAV infection of *Ncbp3⁻/⁻* MEFs compared to *Ncbp3⁺/⁺* controls as seen by a significant distribution shift towards lower induction, which is specific to ISGs and not observed for the total proteome (Fig 4C). Overall, we have identified 282 proteins that change their response to IAV infection upon NCBP3 knockout, including 19 ISGs (Fig 4D, S1 Table). In order to extract functional

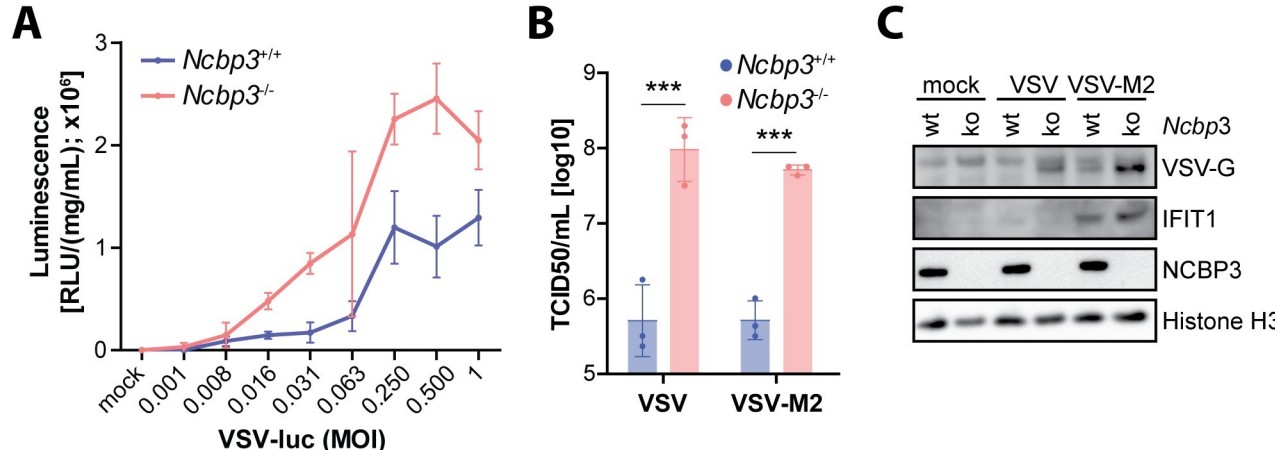

**Fig 3. Loss of NCBP3 supports increased VSV replication *in vitro*.** (A) Luciferase activity in lysate of *Ncbp3* wt and ko MEFs upon infection at different MOIs. MEFs were infected with a VSV-wt reporter virus expressing firefly luciferase using indicated MOIs for 21 hours. Luminescence counts were normalized to total protein concentration. Graphs display the mean ± SD of virus infections performed in triplicates. (B) Viral particle accumulation in supernatants of VSV-infected MEFs quantified by TCID50. MEFs were infected with VSV-wt and -M2 using an MOI of 0.001 for 24 hours. *** $P<0.001$ as analyzed by two-way analysis of variance (ANOVA) with Bonferroni's p-value correction. (C) Western blot analysis of VSV-infected *Ncbp3* wt and ko MEFs using antibodies against indicated proteins. MOI, multiplicity of infection; VSV, Vesicular stomatitis virus; VSV-G, Vesicular stomatitis virus glycoprotein; Ifit1, Interferon-induced protein with tetratricopeptide repeats 1; Ncbp3, Nuclear cap-binding protein 3.

information from observed changes of protein abundances, we performed GO-term enrichment analysis on differentially expressed proteins (S2 Table). Top 10 terms enriched in proteins upregulated upon infection were related to antiviral responses. However, the numbers of proteins associated to individual terms as well as enrichment probability was lower in case of *Ncbp3*-ko MEFs despite overall higher number of differentially expressed proteins (Fig 4E). Among the 19 ISGs, differentially regulated between *Ncbp3*-wt and -ko MEFs upon infection, four types of expression changes could be observed (Fig 4D, green dots, Fig 4F): (I) ISGs that were up-regulated in wt cells after virus infection but showed reduced response in *Ncbp3*-ko: Bst2, Irgm2 and Rtp4; (II) ISGs that lost their up-regulation in *Ncbp3*-ko: Ctnna2, H2-K1, Ifi44, Irgm1, Lgals3bp, Timp2, Ube2l6 and Usp18; (III) ISGs that were up-regulated in wt cells, but became slightly down regulated in *Ncbp3*-ko cells: Helz2 and Tspo; and (IV) ISGs not affected by infection in wt cells and slightly down-regulated in *Ncbp3*-ko cells: Lipa, Gla, Crabp2 and Tor1aip2. Only one ISG, Noc4l, was found to be exclusively upregulated in *Ncbp3*-ko upon IAV infection and not in wt cells. Collectively, these results showed that the induction of ISGs is reduced in the absence of NCBP3, potentially explaining its influence on virus growth.

## NCBP3 is required for cytokine induction and response *in vitro*

Viral infection experiments suggested regulation of innate immune-response genes in an *Ncbp3*-dependent manner. These results prompted us to test MEFs for their ability to respond to the synthetic dsRNA analogue and potent innate immune stimulator polyinosinic-polycytidylic acid (PIC). Remarkably, compared to wt MEFs, PIC-treated *Ncbp3*-deficient cells showed significantly reduced accumulation of IL6 in the supernatants (Fig 5A). This difference was particularly apparent when low amounts of PIC were used and was additionally observed for the NF-κB-induced cytokine RANTES (CCL5) (Fig 5B). Intriguingly, despite significant differences in IL-6 accumulation in the supernatant, *Il6* mRNA expression was similar in both wt and *Ncbp3*-ko MEFs (Fig 5C). These data suggest that cytokine expression requires a

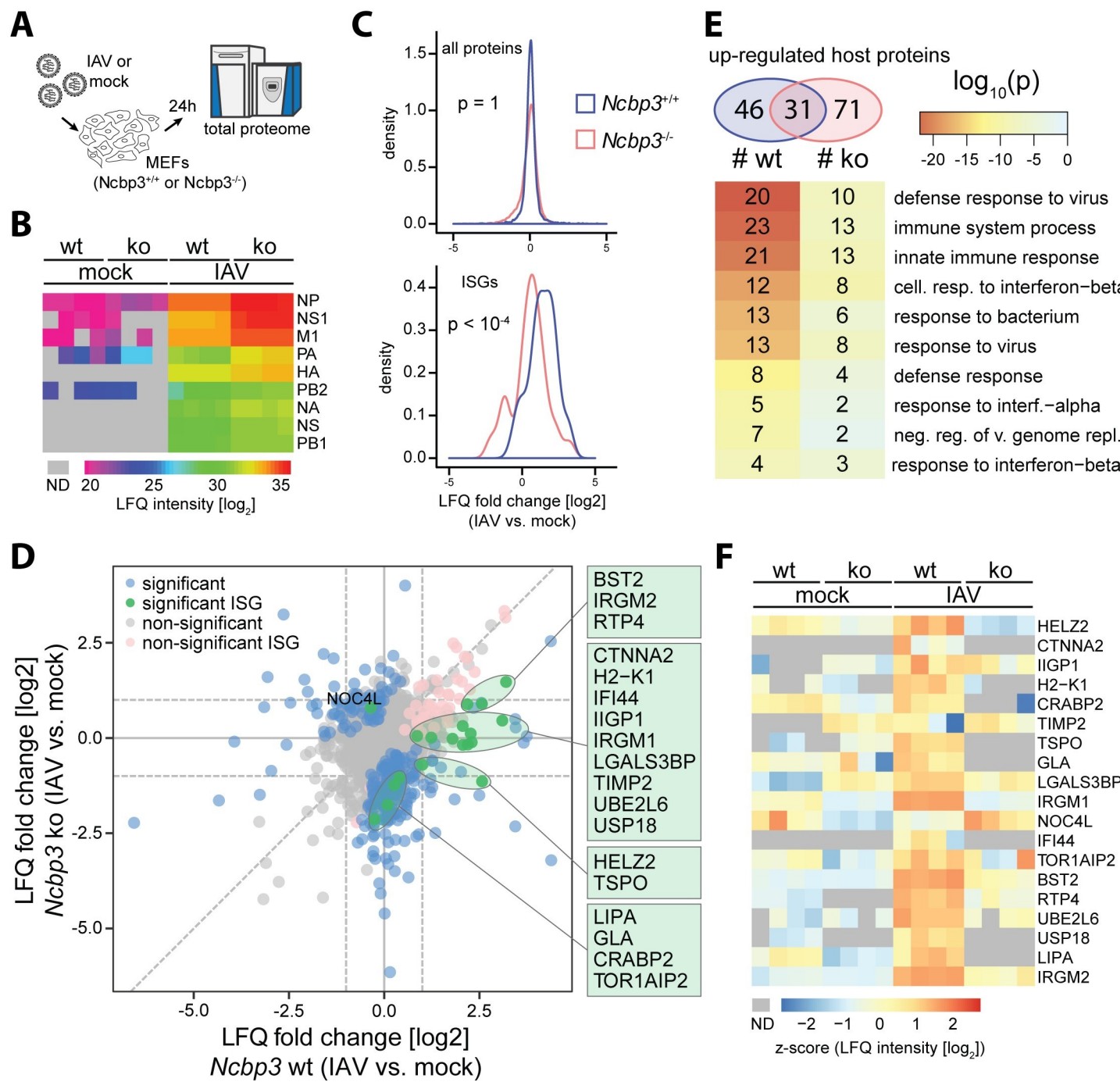

**Fig 4. Global proteomic analyses suggest differential expression of a subset of ISGs.** (A) Design of proteomic experiment. *Ncbp3* wt and ko MEFs were infected for 24 hours with IAV-wt (strain SC35M, MOI 0.1). Cells were lysed and proteomic changes were analyzed by shotgun liquid chromatography-tandem MS (LC-MS/MS). (B) Viral proteins expression in infected MEFs of the indicated genotype. Log2-transformed non-imputed LFQ intensities are shown for each individual replicate. Grey color denotes missing values (ND). (C) Density of $\log_2$ fold change distribution for all quantified proteins (top) or ISGs (bottom) upon IAV infection of *Ncbp3*-wt (blue) and -ko (red) MEFs. P-values were calculated using two-sided Kolmogorov-Smirnov test. (D) Differential expression of host proteins between IAV-infected and mock conditions in Ncbp3 wt (X axis) and Ncbp3 ko (Y axis) MEFs. Proteins exhibiting a significant differential response between the cell lines are highlighted in green (ISGs) or blue (non-ISGs). ISGs that were significantly changing either in *Ncbp3* ko or wt upon infection but do not show significantly different response between *Ncbp3* ko and wt are highlighted in pink. (E) GO-term analysis of up-regulated host proteins upon infection of *Ncbp3* wt and ko MEFs. Numbers of up-regulated proteins is indicated in Venn diagram (top), and top 10 GO-terms ordered by enrichment p-value for infection of *Ncbp3* wt MEFs are shown (bottom). Color of cells correspond to enrichment p-values as indicated in legend and indices in cells correspond to number of protein groups associated to individual GO-terms in respective comparison. (F) Z-scored expression levels of significantly differentially regulated ISGs between infected and mock conditions for *Ncbp3*-ko–wt MEFs. Grey color represents missing values (ND).

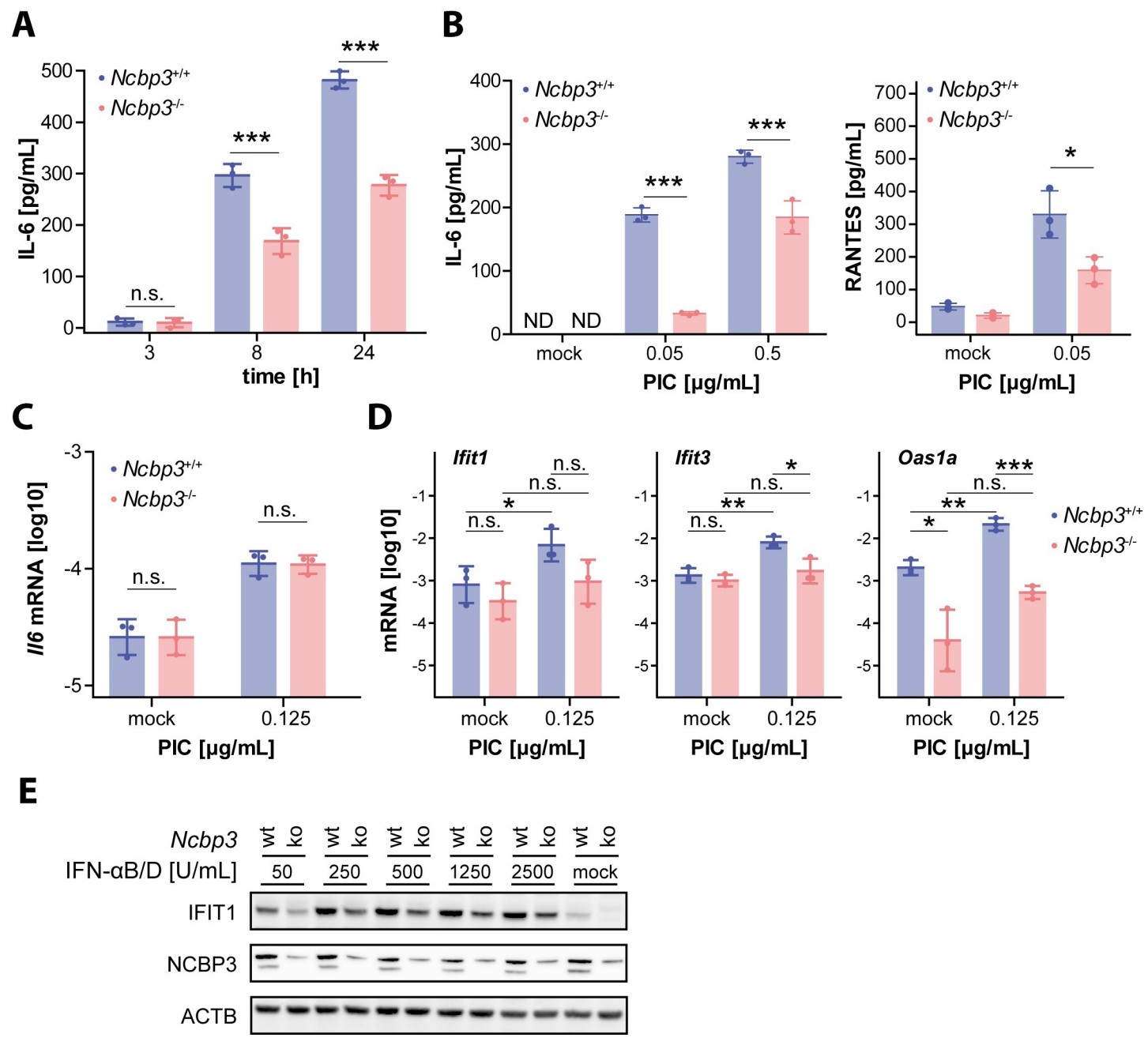

**Fig 5. Ncbp3 depletion alters innate immune responses.** (A-C) Concentration of secreted IL6 and RANTES for *Ncbp3* wt and ko MEFs. MEFs were electroporated with 0.15 (A), 0.5 and 0.05 (B) and 0.125 (C, D) µg/mL PIC and cytokine concentrations in supernatants determined 3, 8 and 24 hours (A) or 24 hours (B) after stimulation by enzyme-linked immunosorbent assay (ELISA). Data represent the individual and the mean value ± SD of PIC treatments performed in triplicates. N.s. $P>0.05$, *** $P<0.001$ as analyzed by two-way analysis of variance (ANOVA) statistics with Bonferroni's post-test. (C) *Il6* mRNA expression in PIC treated *Ncbp3* wt and ko MEFs. MEFs were stimulated with PIC for 16 hours and *Il6* mRNA abundance evaluated by qRT-PCR. Data were normalized to murine *Actb* mRNA and the mean ± SD of three biological replicates consisting of three technical replicates are represented. N.s., $P>0.05$ as analyzed by two-tailed paired sample t-test. (D) Antiviral gene mRNA expression levels in *Ncbp3* wt and ko MEFs, electroporated with 0 (mock) or 0.125 µg/mL PIC. Data represent the mean ± SD values of three biological replicates of stimulation performed in triplicates. * $P<0.05$, ** $P<0.01$, *** $P<0.001$ as analyzed by two-tailed paired sample t-test. *Ifit1*, Interferon-induced protein with tetratricopeptide repeats 1; *Ifit3*, Interferon-induced protein with tetratricopeptide repeats 3; *Oas1a*, 2'-5'-Oligoadenylate Synthetase 1. (E) IFIT1 expression in *Ncbp3* wt and ko MEFs stimulated with IFN. *Ncbp3* wt and ko MEFs were treated with IFN-αB/D for 18 hours. IFIT1 expression was determined by western blot analysis. Depletion of *Ncbp3* and equal loading was confirmed by western blotting against indicated proteins. IL6, interleukin 6; PIC, Polyinosinic:polycytidylic acid; IFN, interferon; NCBP3, Nuclear cap-binding protein 3; IFIT1, Interferon-induced protein with tetratricopeptide repeats 1.

functional alternative CBC that operates on a post-transcriptional level. By extending the repertoire of quantified transcripts, we additionally noticed a clear trend towards lower expression of hallmark ISGs *Ifit1*, *Ifit3* and *Oas1a* in *Ncbp3* ko MEFs upon treatment with PIC (Fig 5D). To test whether NCBP3 is also required to properly execute an interferon response we stimulated MEFs with IFN-αB/D and tested for their ability to synthesize the antiviral protein IFIT1 by western blot analysis. In the absence of NCBP3, IFN-αB/D elicited IFIT1 levels were clearly reduced as compared to those in wt MEFs (Fig 5E). This difference was again particularly evident at low cytokine concentrations and less prominent when high amounts of IFN-α were used for stimulation of cells. Taken together, these data indicated that NCBP3 plays a major role in mounting an antiviral state in response to viral infection, stimulation with a synthetic PRR ligand or treatment with type I interferon and is required for a full-scale antiviral immune response *in vitro*.

### *Ncbp3* deficiency results in increased IAV-induced mortality *in vivo*

Lack of NCBP3 leads to increased virus growth, mitigated cytokine response and reduced induction of a subset of ISGs *in vitro*. To test the relevance of NCBP3 *in vivo*, we infected *Ncbp3*-ko and littermate control mice intranasally with a sublethal dose of IAV. IAV primarily infects the pulmonary epithelium, which exhibits high expression of NCBP3 relative to other tissues (Fig 1B). As expected, control mice survived virus challenge. However, IAV infection resulted in high mortality of *Ncbp3*-deficient mice of which 66.7% had to be euthanized for animal welfare reasons within 13 days of infection (Fig 6A). Histological analysis of IAV-infected wt and *Ncbp3*-ko mice revealed differences in IAV-induced lung pathology between the two genotypes. *Ncbp3*-ko mice showed increased lung inflammatory cell recruitment and pathology characterized by severe epithelial damage and infiltration of lymphocytes, hallmarks of necrotic peribronchial pneumonia caused by influenza A virus infection (Fig 6B and 6C). Collectively, these experiments show that NCBP3 is required to control virus growth and to mount adequate innate immune responses *in vitro* and that loss of NCBP3 results in severe virus-induced inflammation and pathology *in vivo* that correlates with decreased survival.

## Discussion

Regulation of gene expression is one of the most fundamental processes in cellular organisms. To obtain functional proteins, the genetic information has to be transcribed into mRNA and shuttle between the nucleus and the cytoplasm. Coordination of this event is fundamental for precise gene expression in a timely manner. Under challenging conditions such as virus infections, proper functioning of the mRNA processing machinery becomes even more important since an immediate and adequate response needs to be executed. We previously identified human NCBP3 as a cap-binding protein assembling the alternative CBC together with NCBP1 that is involved in the export of polyadenylated RNA from the nucleus in a similar manner as the canonical CBC under physiological conditions [10]. Here, we show that *Ncbp3*-deficient mice are viable, which supports our *in vitro* finding that NCBP3 is dispensable under physiological conditions since it exhibits redundant functions to NCBP2. Surprisingly, breeding of *Ncbp3* heterozygous or knockout mice revealed an atypical Mendelian ratio of born knockout animals suggesting that NCBP3 is required for proper embryonic development, which may be a consequence of its involvement in tightly regulated gene transcription programs that are particularly important during embryogenesis [15].

The alternative CBC is required to respond to specific environmental cues and to allow appropriate gene expression [10], which is critical in coordination of the antiviral response triggered by virus infections [16, 17]. *In vitro*, infections with RNA viruses resulted in elevated

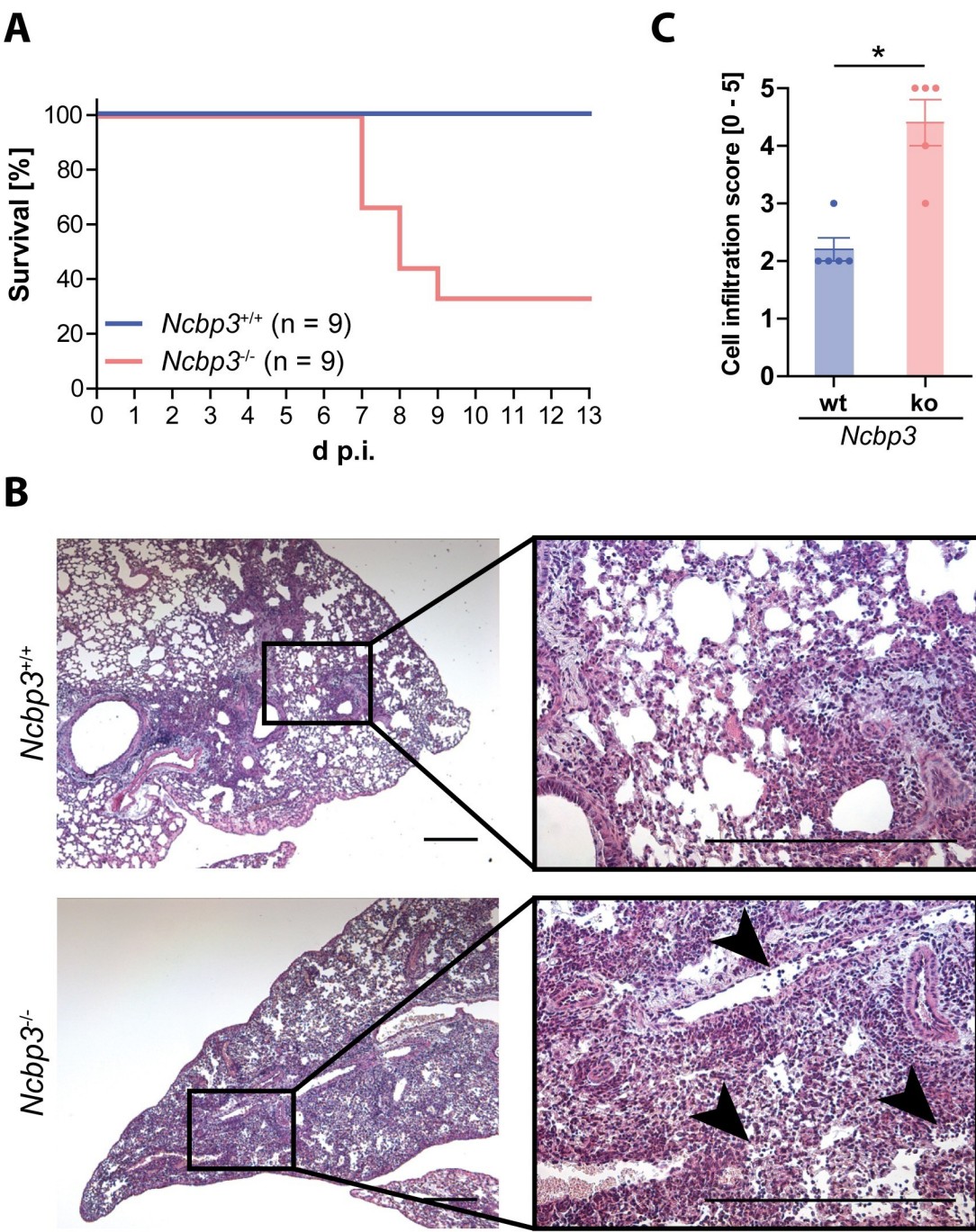

**Fig 6. NCBP3 is required to defend IAV infection *in vivo*.** (A) Survival of *Ncbp3*-wt and -ko mice after infection with IAV-wt (strain SC35M). Mice (n = 9 per group) were infected intranasally with 1,500 PFU of IAV and survival was monitored over 13 days. (B) Histological sections of lungs from IAV-wt infected *Ncbp3*-wt and -ko mice. Mice (n = 5 per group) were infected with 1,500 PFU of IAV for 7 days and paraffin-embedded sections were stained with hematoxylin and eosin. Representative images are shown in 5x (left panel) and 20x (right panel) magnification. Exemplar regions exhibiting prominent lymphocyte infiltration are highlighted (black arrowheads). Scale bar represents 50 μm. (C) Cell infiltration score in *Ncbp3*-wt and -ko lung tissue. Lung sections of 7 days IAV-infected mice (n = 5 per group) were scored for severity of cell infiltration (0–5). Data represent the individual and the mean values ± SD of infiltration scores. * $P<0.05$ as analyzed by one-way analysis of variance (ANOVA) with non-parametric Kruskal-Wallis statistics and Dunn's post-test.

viral titers, increased viral protein production as well as in higher viral transcript expression in *Ncbp3*-deficient cells. Since this was observed both for nuclear- and cytoplasmic-replicating viruses, the effect of NCBP3 appears to rather be on the regulation of antiviral immune responses than a direct interference with viral RNA export and growth. Given the high abundance of viral transcripts and proteins in *Ncbp3*-deficient cells, expression of ISGs, monitored by *Ifit1* and *Ifit3* mRNA, IFIT1 protein and secreted cytokines IP-10 and IL-6, did not scale with the severity of stimulus. This indicates a dysregulation of the normal procession of innate immune response in *Ncbp3*-deficient cells and a general inability of *Ncbp3*-ko cells to translate a stimulatory trigger into protective antiviral immunity. Proteomic analyses of IAV-infected *Ncbp3*-ko and wt MEFs showed reduced expression of a subset of ISGs in *Ncbp3*-deficient cells, including antiviral restriction factors as well as negative regulators of type I interferon, additionally supporting the requirement of NCBP3 for an appropriate antiviral response. Consolidating our data, NCBP3 was necessary for proper expression of antiviral proteins such as IFIT proteins and BST2 (also known as Tetherin), which are antiviral factors active against a wide variety of viruses [3, 18, 19]. While IFIT proteins are nucleic acid-binding proteins that restrict virus replication, BST2 impairs the release of enveloped viruses including IAV and VSV. The impaired induction of restriction factors, such as IFIT proteins and BST2, could explain the higher viral load and elevated accumulation of viral particles in the supernatants of *Ncbp3*-ko compared to wt cells. Similarly, *Ncbp3*-ko MEFs showed reduced expression of USP18 and UBE216, which are involved in ISGylation induced by innate responses. During IAV infection, NS1 is ISGylated which leads to inhibition of viral replication [3, 20, 21]. Alteration in ISGylation in the absence of NCBP3 could therefore further contribute to enhanced viral replication. In sum, our results indicate defects in the induction of an appropriate antiviral immune response in *Ncbp3*-ko cells *in vitro*, which goes in line with higher viral titers.

An involvement of NCBP3 in induction of antiviral immune responses was further supported by stimulation experiments using the synthetic dsRNA analogue PIC and recombinant IFN-αB/D *Ncbp3*-deficient cells showed significantly reduced secretion of pro-inflammatory cytokines IL-6 and RANTES, as well as expression of key ISG IFIT1, particularly when stimulated with low concentrations of ligands. However, at higher concentrations of stimuli, the dependency on NCBP3 gradually decreased. This suggests that when sufficiently high amounts of mRNA are produced, dependency on NCBP3 is alleviated and it is no longer a bottleneck in antiviral protein production. The exact molecular mechanisms of NCBP3 involvement in the antiviral response remain to be elucidated. NCBP3 was previously shown to be sufficient to alleviate the mRNA-export defect induced by depletion of NCBP2 under physiological conditions [10]. The absence of NCBP3 may slow down or reduce the throughput of nuclear to cytoplasm export of specific antiviral transcripts, such as *Il6* mRNA, which could additionally impact the transcripts stability. This theory may explain the reduced post-transcriptional execution of innate immune response observed in NCBP3-deficient cells. Alternatively, cap-independent interactions of NCBP3 with constituents of the mRNA maturation machinery, including the canonical CBC, were recently observed *in vitro*, which could independently or synergistically contribute to the observed phenotype [22]. Such a function of NCBP3 would be predominantly important in the early stages of infection *in vivo*, when little virus stimulus triggers an antiviral response, and unimpaired execution of antiviral programs is required to control initial infection. The exact timing of antiviral responses is of central importance to control virus infection and can dramatically affect the outcome of a disease [23]. Therefore, a fully functional mRNA processing system may be required to allow tight regulation of a rapid response.

In an organism, the antiviral program needs to be strictly coordinated to effectively prevent viral spread and associated pathologies [24, 25]. While timely expression of cytokines and

antiviral proteins is crucial at early stages of infection, expression of negative regulators is essential to prevent overshooting immune reactions after successful virus clearance [16, 26, 27, 28]. This highly dynamic and regulated expression may require the alternative CBC.

## Material and Methods

### Reagents

Hybrid human IFN-αB/D [29], which is highly active on mouse cells, was used for this study. Polyinosinic-polycytidylic acid (PIC; Sigma Aldrich; P9582) and siRNAs (Dharmacon) were transfected into cells using Neon Transfection System (Invitrogen). Primary antibodies were used according to manufacturer recommendation and were as follows: NCBP1 (Thermo Fisher Scientific, PA5-30098), NCBP3 (Atlas Antibodies; HPA008959), Histone H3 (Abcam; ab1791–100). For viral protein detection, we used antibodies against IAV (Millipore; AB1074) and VSV-G (Santa Cruz; sc-66180). Antibodies against NCBP2 were purified from serum of rabbits immunized with recombinant full-length protein purified from E. coli. Serum isolated from *Ifit1*-deficient mice immunized with full-length murine IFIT1 protein purified from *E. coli* was used to detect IFIT1. Antibodies against β-Actin (Santa Cruz; sc-47778) and secondary antibodies detecting mouse (Dako), rabbit (Sigma-Aldrich), goat (Santa Cruz) IgG were horse-radish peroxidase-coupled. CellTiter-Glo Luminescent Cell Viability Assay kit was purchased from Promega. Interleukin-6 was measured by enzyme-linked immunosorbent assay (ELISA) obtained from R&D Systems. Mass spectrometry grade trypsin and LysC was obtained from Wako Chemicals USA and Sigma-Aldrich, respectively. pLIX403 was a gift from David Root (Addgene plasmid # 41395).

### Viruses

All used viruses are classified as BSL2 pathogens in Germany and experiments were carried out according official regulations. Wild-type influenza A virus strain SC35M was used in mouse experiments. In cell culture experiments, we used wild-type as well as NS1-deleted influenza A virus strain SC35M [30] and wild-type and M2-mutated (M51R substitution in M2 protein) vesicular stomatitis virus [31]. IAV-ΔNS1 and VSV-M2 are deficient in their ability to block innate immune responses due to the lack of NS1 and mutation in M2 protein, respectively. In addition, we used an IAV-wt reporter virus [32] that expresses renilla luciferase and a VSV-wt reporter virus expressing firefly luciferase, which were a kind gift from Peter Reuther and Gert Zimmer, respectively.

### Generation of *Ncbp3*-knockout mice

Genetically modified ES cell clones (JM8.F6; C57BL/6N background) carrying a promotor-less targeting cassette to generate "knockout-first" allelic mutation were obtained from the European Conditional Mouse Mutagenesis Program (EUCOMM) [33, 34]. The cassette codes for a neomycin resistance and a LacZ gene with a splice acceptor and a polyA site flanked by FRT sites. Additionally, the second exon of *Ncbp3* is flanked by loxP sites; deletion of exon 2 results in a frame-shift mutation in the *Ncbp3* gene (S1A Fig). Through the insertion of this cassette, a so-called tm1a mutation is generated resulting in the disruption of the targeted *Ncbp3* gene.

Chimeras were generated by ES cell injection into C57BL/6 albino (B6(Cg)-Tyr<c-2J>/J) blastocyst donors, which were implanted into pseudo-pregnant mice. Germ-line male chimeras generated from one ES clone (*Ncbp3* clone A06) were bred with C57BL/6 albino females to produce heterozygous, tm1a-carrying *F1* mice (C57BL/6N-*Ncbp3*tm1a/Mpi). Heterozygous *F1* mice carrying the tm1a mutation in one *Ncbp3* allele were further bred and maintained in the

animal facility of the MPI of Biochemistry at SPF (specific pathogen-free) conditions. Body weight was monitored at weaning (age 19-23 days) and for a group of mice for a period of 13 months. Mouse husbandry was carried out in accordance with animal welfare regulations and have been approved by the responsible authorities (TVA 55.2-1-54-2532-116-2015).

### Genotyping of *Ncbp3* $^{tm1a}$ mice

*Ncbp3*$^{tm1a}$ mice were genotyped using the following primers: Ncbp3_tm1a-fw 5'-CTGTATG TCCGGTCGTCATC-'3, Ncbp3_tm1a-rev 5'-GCCTGCATGTACCATGCATT-'3. PCR conditions were as followed: (1) 94˚C for 1 minute (1×); (2) 94˚C for 30 sec, 57˚C for 20 sec, 72˚C for 30 sec (35×). PCR products were visualized using a 1% agarose gel electrophoresis.

### LacZ/ß-Galactosidase assay

*Ncbp3* expression in tissues was determined by ß-Galactosidase expression driven under endogenous *Ncbp3* promotor from the LacZ gene of the tm1a cassette. Tissues were isolated from heterozygous (*Ncbp3*$^{tm1a+/-}$) and wild-type (background control) mice and homogenized in 2 μL reaction buffer per 1 mg tissue using SS matrix beads (MP Biomedicals) and the FastPrep-24 machine with the following setting: 4x 20 seconds, MP, 4 m/s. Lysates were cleared by centrifugation at 18000x g at 4˚C for 30 minutes. Cleared lysates were used to determine ß-Galactosidase expression using the FluoReporter lacZ/Galactosidase Quantification kit (ThermoFischer Scientific; F-2905) according manufacturer's guidelines. The assay was done in a 384-well format, wherefore, half of the manufacturer's volumes were used. ß-Galactosidase signals were first divided by total protein content in the lysate measured using Pierce 660nm Protein Assay Reagent (ThermoFischer Scientific; 22660) according manufacturer's protocol and then normalized to mean ß-Galactosidase signals in the respective tissue of the same gender.

### Isolation and immortalization of mouse embryonic fibroblasts

Mouse embryonic fibroblasts (MEFs) were generated from day 13.5 embryos from heterozygous intercrosses. All following steps were carried out in a tissue culture hood under sterile conditions. First, uterine horns were dissected out and washed several times in 1x PBS. Embryos were isolated separately from the placenta and its embryonic sac and placed in sterile 1x PBS. Heads, red organs and extremities were removed from the embryos and the remaining tissue placed in fresh 1x PBS. Subsequently, the remaining tissue was minced and incubated for 10 minutes at 37˚C in 3 ml 0.05% trypsin/EDTA (ThermoFischer Scientific). Supernatant containing single cell suspension was cleaned through a cell strainer (Greiner Bio-One International; 542070) adding 30 ml pre-warmed complete DMEM (ThermoFischer Scientific; containing 10% (v/v) fetal bovine serum and antibiotics (100 U/ml penicillin and 100 mg/ml streptomycin)) and subsequently centrifuged at 1000x g for 5 min. Trypsin treatment was repeated several times with remaining tissue. Cells were resuspended in 3 ml complete DMEM and seeded in a cell culture dish. After three passages, the cells were immortalized with SV40-LT by retroviral infection. Immortalized cells were selected with 3 μg/ml puromycin (Sigma-Aldrich; P8833) for eight passages and used for further experiments.

### RNAi-mediated knockdown and cell growth assay

MEFs were electroporated with duplex siRNAs using the Neon Transfection System (Invitrogen) for targeted gene depletion. Duplex siRNAs were obtained from Dharmacon and had the following sequences: Ncbp1 (#1: 5'-AUGCAGAAAUGGACCGAAU-3', #2: 5'-CGUCUGGA CACGAUGAGUA-3', #3: 5'-GGUACGAUGUGAAACGGAU-3', #4: 5'-AGGCCUAACACU

CGCGUUU-3'), Ncbp2 (#1: 5'-CAGCAAAAGUGGUGAUAUA-3', #2: 5'-GCAAUGCGGU ACAUAAACG-3', #3: 5'GUAUGGACGUGGACGGUCU-3', #4: 5'-ACGAGUAUCGGGAG GACUA-3') and scrambled (5'-AAGGTAATTGCGCGTGCAACT-3'). Transfection of siRNA and cell growth assay was previously described elsewhere [10]. Cell growth was analyzed at the indicated time points after the second/repeated siRNA transfection. For western blot analysis, cells were lysed in SSB buffer (62.5 mM Tris-HCl pH 6.8, 2% sodium dodecyl sulfate, 10% glycerol, 50 mM dithiothreitol, 0.01% bromophenol blue) and boiled for 10 minutes at 95°C and subjected to SDS–polyacrylamide gel electrophoresis and western blot analysis.

### Cell growth assays

Cell growth was determined after RNAi-mediated knockdown at indicated time points using CellTiter-Glo (Promega) according to the manufacturer's instructions with the modifications previously described elsewhere [10].

### *In vitro* virus infection

MEFs were seeded on the day before infection and infected with IAV-wt, -ΔNS1 and VSV-wt, -M2. Duration and multiplicity of infection (MOI) is indicated in the figure legends. Supernatants of virus-infected cells were collected and virus titers were quantified by 50% tissue culture-infective dose (TCID50) assays on Vero E6 (purchased from ATCC, CRL-1586) cells. For western blot analysis, cells were lysed in RIPA buffer (50 mM Tris-HCl pH 7.5, 150 mM NaCl, 0.25% sodium deoxycholate, 1% NP-40, 1 mM EDTA) and boiled after addition of Laemmli buffer for 10 minutes at 95°C and subjected to SDS–polyacrylamide gel electrophoresis and western blot analysis. For proteomic analysis, cell pellets were snap-frozen in liquid nitrogen before further processing. For RT-PCR analysis, RNA was isolated using NucleoSpin RNA Plus kit (Macherey Nagel) or Direct-zol RNA MiniPrep Plus (Zymo Research) according manufacturer's protocol.

For luciferase reporter viruses, MEFs were infected with IAV and VSV strains expressing luciferase with different MOIs (indicated in figures). After 21 hours, cells were lysed in 1x passive lysis buffer (Promega) and lysates were mixed with equal volume of 2x renilla reagent solution (100 mM $K_3PO_4$, 500 mM NaCl, 1 mM EDTA, 25 mM Thiourea, 30 μM Coelenterazine) or 2x firefly reagent solution (20 mM Tris-HCl pH 7.8, 0.1 mM EDTA, 3.74 mM magnesium sulfate, 33.3 mM dithiothreitol, 0.27 mM coenzyme A, 0.47 mM D-luciferin, 0.53 mM adenosine triphosphate) in technical duplicates. Subsequently, luminescence was measured using an Infinite 200 PRO series micro plate reader (Tecan). Luminescence counts were normalized to total protein concentration measured using Pierce 660nm Protein Assay Reagent (Thermo Fischer Scientific; 22660).

For reconstitution experiments, *Ncbp3*-ko MEFs were transduced with either *Ncbp3*-bearing or empty lentivirus plasmids. In short, HEK293Ts (a gift from Mary Collins, University College London, UK) grown under standard conditions were transfected with pLIX403-based plasmids containing transgene under doxycycline inducible promoter together with packaging plasmids [35]. At six hours post transfection, growth medium was exchanged. Lentivirus-containing medium was harvested 48 hours post transfection and supplemented to target cells. At 24 hours post transduction, the inoculum was aspirated and replenished with fresh medium or medium containing 1μg/mL doxycycline. Transduced cells were used for further experiments 24 hours post doxycycline induction.

## IFN and PIC treatment

All used concentrations are indicated in figures and figure legends. For IFN treatment, IFN-αB/D was added to the medium of homogenously attached MEFs in a 24-well format and cells were further incubated at 37°C with 5% $CO_2$. Cells were lysed in RIPA buffer (50 mM Tris-HCl pH 7.5, 150 mM NaCl, 0.25% sodium deoxycholate, 1% NP-40, 1 mM EDTA) and boiled in Laemmli buffer for 10 minutes at 95°C and subjected to SDS–polyacrylamide gel electrophoresis and western blot analysis. For PIC stimulation, MEFs were transfected using the Neon Transfection System (Invitrogen) and seeded in 24- or 6-well format. Supernatants were collected after indicated time points and IL-6 concentration was determined using the mouse IL6 DuoSet Elisa (R&D systems) or CCL5/RANTES DuoSet Elisa (R&D systems) according manufacturer's instructions. Cells were lysed and RNA was isolated using NucleoSpin RNA Plus kit (Macherey Nagel) according manufacturer's instructions.

## Quantitative RT-PCR analysis

RNA was reverse transcribed using the PrimeScript RT Reagent Kit with or without gDNA Eraser (Takara/Clontech) and quantified by quantitative RT-PCR using the QuantiFast SYBR Green RT-PCR kit (Qiagen) and a CFX96 Touch Real-Time PCR Detection System (Bio-Rad). Each cycle consisted of 10 seconds at 95°C and 30 seconds at 60°C, followed by melting curve analysis. ΔCt values below -20 were considered below detection range and in corresponding figures marked with not detected (ND). Primer sequences targeting transcripts, originating from indicated mouse genes, were as follows: *Tbp* (5'-CCTTCACCAATGACTCCTATGAC-3'and 5'-CAAGTTTACAGCCAAGATTCA-3'), *Actb* (5'-CTCTGGCTCCTAGCACCATGAAGA-3'and 5'-GTAAAACGCAGCTCAGTAACAGTCCG-3'), *Hmbs* (5'-GAGTCTAGATGGCTCAGATAGCATGC-3'and 5'-CCTACAGACCAGTTAGCGCACATC-3'), *Ifit1* (5'- GGCAGGACAATGTGCAAGAA-3' and 5'- CCATAGCGGAGGTGAATATC-3'), *Ifit3* (5'-TGGTCATGTGCCGTTACAGG-3'and 5'-GCTGCGAGGTCTTCAGACTT-3'), *Ifnb1* (5'-CGGAGAAGATGCAGAAGAGT-3'and 5'-TCAAGTGGAGAGCAGTTGAG-3'), *Tnf* (5'- CAAAATTCGAGTGACAAGCCTG-3' and 5'- GAGATCCATGCCGTTGGC-3'), *Il6* (5'-TAGTCCTTCCTACCCCAATTTCC-3'and 5'-TTGGTCCTTAGCCACTCCTTC-3'), *Oas1a* (5'-CTTCCCCAGGGAGGTACATT-3' and 5'-CTGCATCAGGAGGTGGAGTT-3'), and IAV-*M* (5'-AGATGAGYCTTCTAACCGA-3'and 5'-GCAAAGACATCTTCAAGTYTC-3').

## Quantitative LC-MS/MS-based proteomics

For proteome analysis, four virus infections were performed in parallel. Frozen MEF cell pellets were lysed in U/A buffer (8 M Urea, 100 mM Tris-HCl pH 8.5) shaking for 10 minutes at RT followed by sonication for 15 minutes using a Bioruptor (Diagenode) sonicator on high setting with 30 sec on/off cycles. Lysates were reduced with 10 mM dithiothreitol for 30 minutes at RT and incubated for 20 minutes with 55 mM iodacetamid in the dark to alkylate proteins. Subsequently, a total of thirty micrograms of proteins were digested with LysC (Wako Chemicals USA) and trypsin (Sigma-Aldrich), acidified with TFA and desalted with C18 stage tips. Desalted peptides were analyzed by liquid chromatography coupled to mass spectrometry on a Q Exactive HF MS system (Thermo Fischer Scientific). Raw MS data were processed with MaxQuant version 1.5.5.1 [36] using the built-in Andromeda engine to search against mouse (UniprotKB, mus musculus; Proteome ID UP000000589; release 29/08/2016) and influenza A virus (UniprotKB, strain A/Seal/Massachusetts/1/1980 H7N7; Proteome ID UP000008576; release 27/07/2017) protein sequences (the reverse sequences were used as decoy). Label-free quantification (LFQ) algorithm [37] and Match between Runs option were used with standard settings.

LFQ intensities were log2-transformed and missing values imputed from normal distribution of quantified values using default settings (width of 0.3 and down shift of 1.8). Differential expression analysis was performed using the limma package in R [38]. To identify the proteins, whose response to infection altered between *Ncbp3*-ko and -wt MEFs, the interaction term of the linear model was used. The criterion for significantly changing proteins was |$\log_2$(*fold change*)×$\log_{10}$(*p value*)|>3. ISGs were annotated using the INTERFEROME v2.0 database [39] filtered for mouse fibroblast cells and a minimal fold change of 2. Graphs were plotted in R and adapted using Adobe Illustrator. Z-score was calculated using the following formula: (LFQ(protein)–LFQ(mean of sample)) / sd(sample). Kolmogorov-Smirnov test was performed using R version 3.5.1. GO-term enrichment analysis was performed on differentially expressed host proteins using quantifiable proteome as reference dataset on the level of protein groups using hypergeometric test in R version 3.5.1. GO-terms, associated with more than 2 protein groups and false discovery rate adjusted p-value below 0.01 were considered significant.

## *In vivo* virus infection

6–8 weeks old mice were anesthetized by intra-peritoneal injection of ketamine (100 μg per gram body weight) and xylazine (5 μg per gram body weight). The infection was administered intranasal with 1500 PFU of influenza A virus diluted in 40 μl sterile PBS. Body weight (weight loss) and survival was monitored for 13 days. Mice with a body weight loss greater than 25% of the initial weight were sacrificed and recorded as succumbed to infection. For histology of lung sections, left lobes were isolated from mice, fixed in 4% PFA and embedded in paraffin. Paraffin-embedded sections were stained with hematoxylin and eosin, and cell infiltration scores (0-5) were defined for severity of tissue inflammation as described before [40].

## Ethics Statement

All experiments with mice were carried out in accordance with the guidelines of the Federation for Laboratory Animal Science Associations (FELASA) and the national animal welfare body. Experiments were in compliance with the German animal protection law and were approved by the animal welfare committee of the Regierungspräsidium Freiburg (permit G-12/46).

## Supporting information

**S1 Fig. Characterization of *Ncbp3* knockout mice.** (A) Schematic overview of the promoterless tm1a cassette inserted in intronic *Ncbp3* region. The cassette was inserted into the first intron of *Ncbp3* gene locus (NM_025818.3) flanked by FRT sites and encodes for a neomycin resistance and a LacZ gene with splice acceptor and a polyA site. Exon 2 is flanked by loxP sites which, after recombination, results in a frame-shift mutation. (B) Genotyping PCR of *Ncbp3 tm1a* mice. PCR amplification results in a 324 bp construct for *Ncbp3* wt mice and a 204 bp construct for *Ncbp3* ko (tm1a promoterless cassette insertion) mice. (C) Genotypes of *Ncbp3* chimera crossings. Male *Ncbp3* chimeras were crossed with C57BL/6 albino and obtained genotypes are represented. (D) Genotypes of *Ncbp3 tm1a* crossings. *Ncbp3 tm1a* mice were bred to homogeneity and genotypes obtained for the indicated breeding combination are shown. (E) Body weights of *Ncbp3* wt and ko mice at the age of weaning. Body weight of 52 animals per genotype in the age of 19-23 days were monitored. *** *P<0.001* as analyzed by one-way analysis of variance (ANOVA) statistics with Bonferroni's post-test. (F) Body weight development of *Ncbp3* wt and ko mice over 13 months. Body weight of 6 animals per genotype and gender were monitored for 13 months. FRT, Flipase Recognition Target; EnS 2A, splice acceptor site; T2A, peptide sequence with self-cleaving function; lacZ, lacZ gene encoding for ß-galactosidase; neo, neomycin resistance gene; pA, simian virus 40

polyadenylation signal; loxP, locus of X-over P1; bp, base pair; wt, wild-type; ko, knockout; Ncbp3, Nuclear cap-binding protein 3.
(TIF)

**S1 Table. Quantitative proteomics after IAV infection in Ncbp3 wt and ko MEFs.** Ncbp3 wt and ko MEFs were infected for 24 hours with Influenza A virus (SC35M; MOI 0.1). Proteomic changes were analyzed by quantitative proteomics (see Fig 4 and Methods). Table contains non-imputed label-free quantification (LFQ) intensities of all identified proteins, fold change differences, p-values and pi-scores (pi = -log10(p-value)*logFC) from the linear model comparing infection vs mock for Ncbp3 wt and ko. Interaction fold changes, p-values and pi-scores comparing changes in Ncbp3 wt and ko during infection are depicted as well. Interaction significance is classified into the following categories: (1) sig: proteins showing significant differential regulation between Ncbp3 ko and wt in infection, (2) sig ISG: ISGs with differential regulation between Ncbp3 ko and wt in infection, (3) non-sig: no differential regulation in infection between Ncbp3 ko and wt (4) non-sig ISGs: ISGs with no differential regulation in infection between Ncbp3 ko and wt, but regulated in at least one of the two genotypes between infection and mock (5) viral: viral proteins.
(XLSX)

**S2 Table. GO-terms, enriched for differentially regulated proteins upon infection of MEFs with IAV.** Results of GO-term enrichment of host proteins, significantly changed upon infection of Ncbp3 wt or Ncbp3 ko MEFs or differentially regulated between the infections. Table contains significant GO terms (go_id and go_name), numbers of protein groups associated to go terms in selected comparisons or quantifiable dataset (subset and reference_set, respectively), p-values and false discovery rate adjusted p-values (p and p.adj, respectively), enrichment of proteins associated to GO-term in the subset (enrichment), name of gene ontology (go_name), Ncbp3 genotype of comparison (Ncbp3 genotype) and mode denoting whether GO-term is associated to either up- or down-regulateed proteins in specific comparison. Analysis was performed as described in materials and methods. In short, GO-term was considered significant if number of protein groups associated to it in differentially regulated subset is above 2 and adjusted p value below 0.01.
(XLSX)

## Acknowledgments

We want to acknowledge the innate immunity laboratory for critical discussions and suggestions. Soo-Jin Weißenhorn and the MPI-B transgenic and animal facility for generating and housing mice.

## Author Contributions

**Conceptualization:** Anna Gebhardt, Andreas Pichlmair.

**Data curation:** Anna Gebhardt, Valter Bergant, Andreas Pichlmair.

**Formal analysis:** Arno Meiler, Alexey Stukalov.

**Funding acquisition:** Line S. Reinert, Bernhard Ryffel, Andreas Pichlmair.

**Investigation:** Anna Gebhardt, Valter Bergant, Daniel Schnepf, Dieudonnée Togbe, Claire Mackowiak, Line S. Reinert.

**Methodology:** Anna Gebhardt, Valter Bergant, Daniel Schnepf, Markus Moser, Arno Meiler, Dieudonnée Togbe, Claire Mackowiak, Line S. Reinert, Søren R. Paludan, Bernhard Ryffel, Alexey Stukalov, Peter Staeheli, Andreas Pichlmair.

**Project administration:** Andreas Pichlmair.

**Supervision:** Bernhard Ryffel, Andreas Pichlmair.

**Validation:** Anna Gebhardt, Valter Bergant, Daniel Schnepf, Dieudonnée Togbe, Claire Mackowiak, Line S. Reinert.

**Writing – original draft:** Anna Gebhardt, Andreas Pichlmair.

**Writing – review & editing:** Anna Gebhardt, Valter Bergant, Daniel Schnepf, Markus Moser, Arno Meiler, Dieudonnée Togbe, Claire Mackowiak, Line S. Reinert, Søren R. Paludan, Bernhard Ryffel, Alexey Stukalov, Peter Staeheli, Andreas Pichlmair.

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
