## [Decision Letter · Decision Letter 0]

29 Jul 2019

Dear Dr Pichlmair:

Thank you very much for submitting your manuscript "The alternative cap-binding complex is required for antiviral defense in vivo " (PPATHOGENS-D-19-01246) for review by PLOS Pathogens. Your manuscript was fully evaluated at the editorial level and by independent peer reviewers. The reviewers appreciated the attention to an important topic but identified some aspects of the manuscript that should be improved. In particular, reformat the manuscript as suggested to make it more comprehensive, and study the abundances of few more selected ISGs to solidify the data.

We therefore ask you to modify the manuscript according to the review recommendations before we can consider your manuscript for acceptance. Your revisions should address the specific points made by each reviewer.

(1) A letter containing a detailed list of your responses to the review comments and a description of the changes you have made in the manuscript. Please note while forming your response, if your article is accepted, you may have the opportunity to make the peer review history publicly available. The record will include editor decision letters (with reviews) and your responses to reviewer comments. If eligible, we will contact you to opt in or out.

(2) Two versions of the manuscript: one with either highlights or tracked changes denoting where the text has been changed; the other a clean version (uploaded as the manuscript file).

We hope to receive your revised manuscript within 60 days or less. If you anticipate any delay in its return, we ask that you let us know the expected resubmission date by replying to this email.

[LINK]

Sincerely,

Peter Sarnow

Guest Editor

PLOS Pathogens

Sara Cherry

Section Editor

PLOS Pathogens

Kasturi Haldar

Editor-in-Chief

PLOS Pathogens

orcid.org/0000-0001-5065-158X

Grant McFadden

Editor-in-Chief

PLOS Pathogens

orcid.org/0000-0002-2556-3526

Reviewer's Responses to Questions

**Part I - Summary**

Reviewer #1: In this work, the authors look at the role of NCBP3 alternative cap binding complex during viral infection. They show that the loss of this protein leads to a decoupling of transcription and translation of antiviral genes, which increases susceptibility to viral infection. The data are generally of high quality and I think the story is compelling an on an important topic.

Reviewer #2: Coordinated regulation of gene expression is essential for a proper innate immune response in response to infection. This can happen at any stage of gene expression, including transcription, mRNA stability and translation. Previously, Gebhardt et al discovered that an alternative cap-binding complex, consisting of NCBP1 and NCBP3, which is important during cell stress like VSV, EMCV, and SFV virus infection. Following upon this first paper, here Gebhardt et al determine the effect of NCBP3 knockout on Influenza A infection. They generate NCBP3 knockout mice, and show that in knockout cells IAV replicates to a higher titer compared to wild type cells. This increased replication correlates with decreased Ifnb1 mRNA levels, suggesting that NCBP3 is important for the cell to establish an antiviral state. Notably, mRNA levels of another ISG, Ifit3, are unchanged, indicating there is specificity to the regulation by NCBP3. To determine the proteome dependent on NCBP3, Gebhardt et al perform proteomics in IAV-infected WT and NCBP3 knockout cells. They discover that ~5% of the detected proteome is altered upon knockdown, including 19 ISGs. In agreement to the importance of NCBP3 in establishing an antiviral state, cells do not correctly respond to interferon or PIC treatment, and knockout mice have significantly reduced survival rates upon challenge with IAV.

Overall, this is a very nice paper that implicates NCBP3 as an important player in the antiviral response. I have minor specific suggestions and questions about some discrepencies in the data:

1. Fig 1: Where is the western blot for the efficiency of NCBP1 knockdown by siRNA?

2. Fig 4B: Why are peptides for IAV proteins detected in the mock infected samples?

3. Fig 4: Please add gene ontology analysis of all 282 proteins whose expression is changed upon NCBP3 knockdown. Are there other stress response pathways being altered besides the ISG response?

4. Fig 4: In the prior paper discovering NCBP3, Gebhardet et al performed CLIP with NCBP3 – what population of mRNAs that crosslinked to NCBP3 encode proteins that are altered in the proteomics experiment? Does enrichment of RNA binding correlate with the level of change in protein levels upon knockdown?

5. Fig 2/5: In Fig 2, IAV infection induces the same amount of IFIT1 protein in both wild type and mutant cells, suggesting that NCBP3 is not important for the increase in IFIT1 levels upon infection. However, in Fig 5, IFIT1 protein levels are reduced in NCBP3 knockout cells upon IFN treatment. Why is there this discrepancy?

In addition, these data do not agree with the text on page 13/14, which should be reworded: “…expression of ISGs, monitored by Ifit3 mRNA and IFIT1 protein, did not scale with severity.” - in Fig 2C the levels of Ifit3 mRNA are identical between knockout and wild type cells, and the levels of IFIT1 protein also are the same in both cell types. In addition, despite in the SI table, IFIT1 is a “non-sig ISG,” on p14/15, Gebhardt et al speculate that Ifit1 mRNA stability is affected by NCBP3. This section should also be reworded.

Reviewer #3: The submitted manuscript by Gebhardt et al. reports the role of NCBP3, which along with NCBP1 forms an alternative cap-binding complex (CBC), in antiviral defense against RNA viruses in vivo. Through a series of ex vivo-, proteomic-, and in vivo-based approaches, the authors determine that NCBP3 is necessary to mount an antiviral response, and there is a unique subset of ISGs that are significantly regulated by NCBP3. The authors use MEFs from Ncbp3+/+ and Ncbp3-/- mice to demonstrate lack of NCBP3 results in enhanced virus growth and a dampened antiviral response. These data are corroborated by survival studies and histology in vivo.

This manuscript is an extension of the work done by the same group published in Nature Communications in 2015, in which the authors nicely describe an alternative CBC in which NCBP2, the canonical nuclear cap-binding protein can compensate if NCBP3 expression is lost, and that NCBP3 is important under conditions of virus infection. The current manuscript significantly expands on this idea by moving to an in vivo model (newly generated Ncbp3-/- mice ) and by utilizing several different viruses while also investigating in greater detail the antiviral response. Overall, the experiments are well-executed and the data convincing. There are only a few remaining points that the authors should address in order for this manuscript to be considered for publication:

**Part II – Major Issues: Key Experiments Required for Acceptance**

Reviewer #1: My major critique is that the figures appear somewhat disjointed and the message of the report becomes jumbled. As I interpret the data, the key message is that loss of NCBP3 affects translation of certain antiviral genes. The proteomic data shows this at the protein level for a subset of ISGs, and for some genes, the authors show that RNA levels aren’t affected. There however, is no systematic presentation of the data. Sometimes they show RNA of one gene, sometimes protein from another, and in essentially all cases, it is not clear why a specific gene is selected (especially of the IL6 data). Furthermore, none of the validation data appears to be linked to the proteomic data, which is a centerpiece of the paper.

Some specific comments for the figures are below. But overall, I believe the impact of the paper would be significantly strengthened with some reformatting. If the report showed a systematic determination of which antiviral genes had defects specifically in translation after the loss of NCBP3 (maybe RNAseq combined with the proteomic data), and then those genes were validated in vitro, the story would be much more digestible. As currently presented, the genes individually tested before (and after) the proteomics don’t appear to come out as hits, and it is difficult to understand why the specific genes tested with qPCR and western were chosen (or how they relate to the in vivo phenotype). The phenotypes are clear, and the in vivo data are striking, but it is currently difficult to understand the scope and magnitude of the NCBP3 effects on antiviral gene expression.

Major Comments:

Figure 2C: This analysis should have statistical significance indicated, and the authors are concluding sometimes the knockout significantly changes things and sometimes not.

Figure 2C,D: Why was IFIT3 quantified at the RNA level and IFIT1 at the protein level? Do different ISGs all behave the same way? Showing several ISGs at both the RNA and protein level would be informative.

Figures 2,3: The interpretation of antiviral gene expression is somewhat complicated by differences in viral replication. Transfecting PolyIC may be a “cleaner” way to quantify antiviral response in the absence of effects of viral replication. The knockout phenotype may be more apparent in this system.

Figure 4: It would be nice to have matched RNA levels for this experiment, this could support the data from figure 2.

Figure 5: Why was IL6 selected? There is no clear link to IL6 from any of the previous experiments that I could find.

Reviewer #2: Comment 5 from the above suggestions needs to be addressed, specifically the discrepencies between the different data needs to be addressed, or the Ifit mRNAs should not be highlighted in the discussion as NCBP3 targets.

Reviewer #3: Figures 2C/3C. Additional ISG genes should be measured in order to characterize better the IFN/ISG response in wt vs knockout cells. Since the authors describe a subset of differentially regulated ISGs in Figure 4, these would also be nice to test and validate by qRT-PCR.

Figure 5. Similarly to the point above, it would be nice if the authors could test additional genes to make a better generalization of the antiviral activity of NCBP3. Additional chemokines (like CCL5 tested by ELISA), type I IFN (IFNB1) or ISGs (IFIT1, IFIT3, MX1, OAS), and particularly the ones identified by the proteomic analysis, would be nice to show and supplement the current data.

To confirm that NCBP3 is controlling the antiviral effect, Ncbp3-/- MEFs should be reconstituted with NCBP3, followed by virus infection.

Figure 6. What is the effect on viral titers (or viral mRNA) in wt vs knockout mice, e.g. in the lungs? Is there any differential cytokine and ISG regulation in these mice?

**Part III – Minor Issues: Editorial and Data Presentation Modifications**

Reviewer #1: The light blue and light red colors are sometimes difficult to distinguish, perhaps darker variants of each color would help with clarity.

Reviewer #2: See comments 1-4 in summary.

Reviewer #3: Figure 2C. Neither error bars nor significance are provided for these qRT-PCR data.

Figure 4. Could the authors describe how ISGs were grouped and defined?

Figure 4D. It is unclear to the reviewer what the difference is between the pink (non-significant ISG) and green (significant ISG) as the text states, “Overall, we have identified 282 proteins that change their response to IAV infection upon NCBP3 knockout, including 19 ISGs (Fig 4D, S1 Table).” However, from the figure, there seem to be pink dots overlapping with green ones, which appears that significance would be quite similar.

Figure 6B. Arrowheads could be used to make areas of lymphocyte infiltration clear in the slides.

PLOS authors have the option to publish the peer review history of their article (what does this mean?). If published, this will include your full peer review and any attached files.

Reviewer #1: No

Reviewer #2: No

Reviewer #3: No

---

## [Editor Report · Decision Letter 1]

23 Oct 2019

Dear Dr Pichlmair,

We are pleased to inform that your manuscript, "The alternative cap-binding complex is required for antiviral defense in vivo", has been editorially accepted for publication at PLOS Pathogens. 

Before your manuscript can be formally accepted and sent to production, you will need to complete our formatting changes, which you will receive by email within a week. Please note that your manuscript will not be scheduled for publication until you have made the required changes.

IMPORTANT NOTES

(1) Please note, once your paper is accepted, an uncorrected proof of your manuscript will be published online ahead of the final version, unless you’ve already opted out via the online submission form. If, for any reason, you do not want an earlier version of your manuscript published online or are unsure if you have already indicated as such, please let the journal staff know immediately at plospathogens@plos.org.

(2) Copyediting and Proofreading: The corresponding author will receive a typeset proof for review, to ensure errors have not been introduced during production. Please review the PDF proof of your manuscript carefully, as this is the last chance to correct any errors. Please note that major changes, or those which affect the scientific understanding of the work, will likely cause delays to the publication date of your manuscript. 

(3) Appropriate Figure Files: Please remove all name and figure # text from your figure files. Please also take this time to check that your figures are of high resolution, which will improve the readbility of your figures and help expedite your manuscript's publication. Please note that figures must have been originally created at 300dpi or higher. Do not manually increase the resolution of your files. For instructions on how to properly obtain high quality images, please review our Figure Guidelines, with examples at: http://journals.plos.org/plospathogens/s/figures.

(4) Striking Image: Please upload a striking still image to accompany your article if one is available (you can include a new image or an existing one from within your manuscript). Should your paper be accepted, this image will be considered for our monthly issue image and may also appear on our website to feature your article. Please upload this as a separate file, selecting "striking image" as the file type upon upload. Please also include a separate "Other" file with a caption, including credits and any potential copyright information. Please do not include the caption in the main article file. If your image is from someone other than yourself, please ensure that the artist has read and agreed to the terms and conditions of the Creative Commons Attribution License at http://journals.plos.org/plospathogens/s/content-license. Please note that PLOS cannot publish copyrighted images.

(5) Press Release or Related Media: If your institution or institutions have a press office, please notify them about your upcoming paper at this point, to enable them to help maximize its impact. If they will be preparing press materials for this manuscript, please inform our press team in advance at plospathogens@plos.org as soon as possible. We ask that you contact us within one week to plan ahead of our fast Production schedule. If you need to know your paper's publication date for related media purposes, you must coordinate with our press team, and your manuscript will remain under a strict press embargo until the publication date and time. This means an early version of your manuscript will not be published ahead of your final version. 

(6)  PLOS requires an ORCID iD for all corresponding authors on papers submitted after December 6th, 2016. Please ensure that you have an ORCID iD and that it is validated in Editorial Manager.  To do this, go to ‘Update my Information’ (in the upper left-hand corner of the main menu), and click on the Fetch/Validate link next to the ORCID field.  This will take you to the ORCID site and allow you to create a new iD or authenticate a pre-existing iD in Editorial Manager

(7) Update your Profile Information: Now that your manuscript has been provisionally accepted, please log into Editorial Manager and update your profile, if needed. Go to https://www.editorialmanager.com/ppathogens, log in, and click on the "Update My Information" link at the top of the page. Please update your user information to ensure an efficient production and billing process. 

(8) LaTeX users only: Our staff will ask you to upload a TEX file in addition to the PDF before the paper can be sent to typesetting, so please carefully review our Latex Guidelines http://journals.plos.org/plospathogens/s/latex in the meantime.

(9) If you have associated protocols in protocols.io, please ensure that you make them public before publication to guarantee immediate access to the methodological details.

Best regards,

Peter Sarnow

Guest Editor

PLOS Pathogens

Sara Cherry

Section Editor

PLOS Pathogens

Kasturi Haldar

Editor-in-Chief

PLOS Pathogens

orcid.org/0000-0001-5065-158X

Grant McFadden

Editor-in-Chief

PLOS Pathogens

orcid.org/0000-0002-2556-3526
---

## [Editor Report · Acceptance letter]

12 Dec 2019

Dear Dr Pichlmair,

We are delighted to inform you that your manuscript, "The alternative cap-binding complex is required for antiviral defense *in vivo*," has been formally accepted for publication in PLOS Pathogens.

Best regards,

Kasturi Haldar

Editor-in-Chief

PLOS Pathogens

orcid.org/0000-0001-5065-158X

Grant McFadden

Editor-in-Chief

PLOS Pathogens

orcid.org/0000-0002-2556-3526